# Rigid Polyurethane Biofoams Filled with Chemically Compatible Fruit Peels

**DOI:** 10.3390/polym14214526

**Published:** 2022-10-26

**Authors:** Andrey Pereira Acosta, Caio Gomide Otoni, André Luiz Missio, Sandro Campos Amico, Rafael de Avila Delucis

**Affiliations:** 1Postgraduate Program in Mining, Metallurgical and Materials Engineering, Federal University of Rio Grande do Sul, Porto Alegre 91509-900, Brazil; 2Department of Materials Engineering (DEMa), Federal University of Sāo Carlos (UFSCar), Sāo Carlos 13565-905, Brazil; 3Postgraduate Program in Materials Science and Engineering, Federal University of Pelotas, Pelotas 96010-610, Brazil

**Keywords:** bergamot peel, banana peel, food waste, reinforced expansive foam

## Abstract

Banana and bergamot peels are underutilized byproducts of the essential oil and juice-processing industry. This study was designed for the development of rigid polyurethane foam (RPUF) composites using polysaccharide-rich fruit peels as fillers. These fillers were characterized for chemical properties using wet analyses. Additionally, the influences of the filler type and filler content on morphological, thermal, mechanical, hygroscopic, and colorimetric properties of the RPUF were investigated. The main results indicated that, in a comparison with the neat RPUF, the insertion of up to 15% of fillers yielded similar water uptake, apparent density, compressive strength, and color properties, as well as increases up to 115% in thermal stability and up to 80% in cell size.

## 1. Introduction

The worldwide consumption of polyurethane (PU) parts is above USD 60 billion and it is forecasted to reach more than USD 80 billion in coming years [1]. PU foams are among the most popular thermosetting plastics produced from petroleum. These cellular plastics are divided into three categories, namely rigid, semi-rigid and flexible ones, which depend on their crosslinking degree and density. Rigid PU foams (RPUF) are valuable insulation materials due to their closed-cell structure, low thermal conductivity, high compressive strength, and low water absorption [2].

RPUF with densities of at least 40 kg/m^3^ are commonly used in insulation, automotive parts, refrigeration, building engineering, and packaging [3]. They present similar properties if compared to some other traditional insulation materials, such as expanded polystyrene, extruded polystyrene, porous concrete, cork tile, and mineral wool [4]. Although the thermal insulation properties of RPUF are better than lots of insulation materials, the use of RPUF is sometimes not preferred in some applications because of their high cost [5].

In addition to that, it is also estimated that the oil will run out in ~50 years and, therefore, the replacement of petroleum-based raw materials by renewable resources is one of the greatest global challenges [2]. According to Członka and coworkers [1], the dependence on petrochemical products is the main problem associated with the PU industry, which includes isocyanates, polyols, and several additives (e.g., surfactants, flame retardants, catalysts, chain extenders). Therefore, cheap and bio-based fillers may play a significant role in the future of the RPUF industry, reducing both raw material and production costs. Additionally, the environmental appeal of these RPUFs can be improved by the insertion of discredited vegetable materials or residues. This trend is also currently followed by the industry of several other polymer materials, such as polypropylene, polylactide, poly(trimethylene carbonate), poly(butylene succinate), polyvinyl chloride, and many other polymers [6].

Cellulose-based fibers, micro/nanofibers, nanocrystals, paper and powder, wood flour, lignin particles, natural macro-fibers, silica-based micro/nanoparticles, and domestic and industrial solid residues are some already studied fillers for RPUF [7,8,9,10]. These vegetable resources provide a higher cost benefit compared to synthetic fibers and also represent an increase in sustainable appeal [11]. The main advantages conferred by the introduction of natural fillers into RPUF include increases in thermal, mechanical, and hygroscopic properties and biodegradability, as well as decreases in consumption of oil-based raw materials, emissions of greenhouse gas, and production costs [5]. Most of these improvements in technological properties imparted by bio-based fillers are associated with the presence of free hydroxyl (OH) groups on their surface, which are prone to form urethane linkages with isocyanate (NCO) groups from the PU backbone [12,13]. Therefore, high contents of cellulose and hemicelluloses in the fillers can be considered indicators of high chemical affinity with a PU system [7].

In this sense, biomass wastes from food processing are one of the most abundant sources of renewable raw materials [6] and most of them were not addressed yet as fillers for RPUF. From an environmental perspective, it is important to reuse fruit peels and explore them as potential raw materials for producing value-added products. This would help to reduce the burden on ecology and contribute to the world economy. In this sense, among the fruit peels, banana and bergamot present two of the highest holocellulose (cellulose + hemicelluloses) contents and, because of that, may present high host compatibilities with PU systems.

Banana (Musa species) is a tropical fruit harvested throughout the year. Banana trees widely grow in different warm countries and this fruit is an important part of the diet in many tropical populations. Indeed, this is the world’s second largest fruit crop in consumption with more than 100,000,000 tons per year [14]. In addition to the direct consumption, this fruit is used for several food processing industries, including the production of dried pulps, jam, chips, spirits distilled from wine/beer [15]. Studies on maximizing the use of this fruit have been addressing new applications for leaves, pseudo stem, stalk, and peels. The banana peel represents 30–40% of the entire banana weight, generating about 50 million tons of waste annually, so its peel wastes are readily available and underutilized [8]. Bergamot fruits are mainly processed by wash-scraping for extract juice and essential oils. Their peels can be used for producing animal food and represent 50–65% of its total weight [16].

Although there is an increasing demand for both these fruits, the huge amount of waste generated actually has given rise to several problems related to solid-waste management and safe disposal. Landfills have been the most common method of waste disposal, but in some cases, burning in open is preferred [15]. Therefore, in this study three different proportions (5, 10, and 15%) of banana and bergamot peels were incorporated in RPUF.

## 2. Materials and Methods

### 2.1. Materials 

Banana (*Musa* sp.) and bergamot (*Citrus bergamia*) fruits were acquired from a supermarket located in Pelotas/Brazil. After consuming the pulps, the fruit peels were oven-dried (at 50 °C until reaching constant mass) and ground in a Wiley mill (Mark Marconi Model MA340, São Paulo/Brazil) coupled to a 100-mesh screen (≤150 mm). These dried fillers were prepared for wet chemical analyses (Tappi 257 cm-12) and then ashes (T211 om-93), ethanol–toluene extractives (Tappi T204 om-97), acid-insoluble (Klason) lignin (Tappi T222 om-98), and holocellulose (remaining mass up to 100%) contents were determined. Castor oil, glycerin, polyethylene glycol (PEG-400) (chain extender), silicon oil (surfactant), and dimethylbenzylamine (catalyst) were purchased from G. Gotuzzo & Cia. Ltd. (Pelotas/Brazil). The polyol was a 3:1 weight ratio mixture of castor oil and glycerin. Isotane DM, which is a polymeric methylene diphenyl diisocyanate (p-MDI), was purchased by Polisystem Indústria e Comércio de Poliuretano (Porto Alegre/Brazil) and used as an NCO source. Other details on these reagents can be consulted in a previous study of the group [7].

### 2.2. RPUF Manufacture 

RPUF parts were produced by a single-shot method using two components (A and B) adjusting an NCO/OH ratio of 1.2. Component A (polyol mixture), consisting of a mixture of castor oil (67.5 parts/g), distilled water (4 parts/g), glycerol (22.5 parts/g), PEG-400 (10 parts/g), silicon oil (2.5 parts/g) and filler, was mechanically stirred at 1000 rpm for 60 s using Fisatom equipment (São Paulo/Brazil). Later, component B (p-MDI and amine (1 part/g)) was added to component A, which kept being stirred for extra 120 s. All RPUF freely rose inside opened wood boxes. Figure 1 summarizes the synthesis of the RPUF parts. After full rising, the RPUF was left to cure for 2 h at 60 °C and post-cure for two weeks at room temperature, as suggested by Delucis and co-workers [7].

### 2.3. RPUF Characterization

Chemical groups from the post-cured RPUF were analyzed using a Fourier transform infrared spectrophotometer (FTIR) in IRSpirit equipment (Shimadzu^®^ brand, Quioto/Japan) equipped with a diamond attenuated total reflection (ATR) accessory. The spectra were recorded on the reflectance mode in the wavenumber range of 500–4000 cm^−1^ over 64 scans with a resolution of 4 cm^−1^.

Morphology of the RPUF was analyzed perpendicular to the rise direction by scanning electron microscopy (SEM) in MA10 equipment (Zeiss Evo brand, Oberkochen/Germany) operating at 3 kV. Average cell width, length, and anisotropy index were measured using ImageJ software (Version 1.53t 24), as described by Delucis and coworkers [17].

Thermal stability of 10 mg RPUF samples was evaluated using a TGA-1000 thermogravimetric analyzer (Navas brand, Conway/AR/USA), which was heated from 25 to 800 °C at a heating rate of 20 °C/min under a nitrogen atmosphere by purging 50 mL/min gas.

Apparent density of seven prismatic RPUF samples per group (5.0 cm × 5.0 cm × 2.5 cm) was taken using an analytical scale (0.001 g resolution) and a digital caliper (0.01 mm resolution) in 5 samples. These same specimens were tested under compression load parallel to the rise direction in a 23-5D universal testing machine (Emic brand, Norwood/MA/USA). The crosshead speed was 2.5 mm·min^−1^ and the compressive strength was read at a strain deformation of 13%, according to ASTM D1621 [18].

Water uptake was also evaluated for the RPUF in 5 samples. This property was determined as the weight percentage gain after 24 h of water immersion. Prior to each weighting, the excess water was retired using filter paper. This method followed ISO 2896 [19], except for the dimensions, which were 5.0 cm × 5.0 cm × 2.5 cm. 

The effect of the fillers in the RPUF’s color was evaluated using a CR-400 colorimeter (Konica Minolta brand, Chiyoda/Japan), which provided brightness (L*), green–red (a*) and blue–yellow (b*) coordinates, as well as Chroma (C*) and hue angle (h). The apparatus was configured to use a light source (illuminant) D65 and 10° viewing angle, using the method known as CIELab.

All data, except the chemical and water uptake results, were subjected to ANOVA tests. Whenever the null hypothesis was rejected, Tukey tests were used to compare the means. Before that, homogeneity of variances and data normality were verified using Leven and Shapiro–Wilk tests, respectively. All statistical analyses were implemented at a significance level of 5%.

## 3. Results and Discussion

As expected, both fruit peels presented high holocellulose contents (above 60%), [20,21] which confirms their high host compatibility with the PU system [22] (Figure 2). The bergamot peel was slightly superior if compared to the banana one in terms of holocellulose content. The polysaccharides from banana peels include cellulose, pectin, sucrose, glucose, and fructose, among others [15,23]. According to [16], apart from the cellulose, bergamot peels present some different hemicelluloses, such as rhamnose, fucose, arabinose, xylose, mannose, galactose, glucose, and galacturonic acid. The opposite trend was found for the lignin content, in which the banana peel showed that it is a more lignified tissue compared to the bergamot. The banana peel also presented a higher ashes content if compared to the bergamot one, which can be explained by its high lignin content, which partly becomes fixed carbon under pyrolytic processes [7].

All the RPUF presented similarly shaped FTIR spectra (Figure 3), which indirectly indicates chemically similar structures. Main FTIR signals were found at 900–1200 (C–O–C groups from the polyol and C–N urethane linkages), 1520 (N–H amide II groups), 1720 (H–C=O urethane linkages) and 2850–2970 (CH_2_ and CH_3_ groups) cm^−1^ [8,12,24]. In addition, the broad peak at 3400 cm^−1^ can be associated with –OH groups from absorbed moisture [8]. The prominent peak at 2270 cm^−1^, which was founded for the RPUF filled with 10 and 15 wt% of bergamot, is related to unreacted NCO groups. This indicates that NCO groups from the p-MDI probably were entrapped in the cellular structure of this RPUF due to particular nucleation mechanisms imparted by this filler. This unclear nucleation process may be attributed to the high content of carbohydrates in the bergamot peel. In a comparison between the bergamot-filled RPUF and the neat one, a peak at 1670 cm^−1^ appears in the former one. According to Bossa and coworkers [9], this FTIR event indicates a decrease in urethane density (decrease in H-C=O in urethane linkages), accompanied of formation new flexible domains, such as urethane (f-C=O) and urea (f-C=O) chains. This occurs because NCO groups from the p-MDI may form covalent bonds with OH groups from the filler instead of reacting with OH from the polyol.

Most cells from both the neat and filled RPUF look closed and rounded, except the bergamot-filled RPUF with 10% and 15% filler contents. The SEM micrographs (Figure 4) also suggest a good chemical affinity of the fillers in the RPUF cellular structure since no agglomerates were founded. Similar behaviors have been found in the literature for other chemically compatible fillers, such as fillers based on different polysaccharides [5,11,25]. These later PU systems again seem to be affected by urethane linkages between NCO groups from the p-MDI and OH groups from the bergamot peel. These reactions may become nucleation sites. The ability of some fillers of becoming nucleating centers was described by Kairytė and coworkers [26]. According to these authors, the filler attaches itself to the cell wall, encapsulating blowing gas, and then becoming a nucleation site, which can weaken and even break up cell edges, due to the weak interaction between NCO groups and OH groups of the filler.

The morphological properties of the RPUF confirmed some characteristics visualized in the SEM images, in which similar cell length and cell width explain the found rounded cells (Table 1). Only the RPUF filled with 15% of bergamot peels was statistically higher than the neat one in terms of cell length and anisotropy index. In a comparison with the other PU systems, this characteristic attributed to the bergamot-filled RPUF at 15% indicates that nucleation mechanisms ascribed to filler/RPUF reactions yielded greater and more elliptical cells oriented in the rise direction, which may be positive for both thermal and mechanical performances [27]. On the other hand, all RPUF incorporated with filler contents up to 10% presented similar cell bubbles.

In all, there was no difference ascribed to the filler content, which can be explained by a slightly uneven distribution of the filler in the RPUF matrix since the sample prepared for this analysis is too small. Therefore, the different crosslinking densities indicated by wet chemical and FTIR results were not detected by TG analysis (Figure 5). Characteristic temperatures were defined as T2% (temperature attributed to 2% of weight loss), T5% (temperature attributed to 5% of weight loss), and T50% (temperature attributed to 50% of weight loss) (Table 2).

The release of volatile products from the neat RPUF increased with the incorporation of both fillers at all contents as found by the increase in T2%. Considering that these fillers presented high volatile matter contents, more prone to be released at lower temperatures, the higher T2% values were attributed to their partial crosslinking with the NCO groups. Furthermore, the filled RPUF presented slower initial thermal degradation than the neat one in all cases, with T5% in the 95–250 °C range, which is also probably due to filler/RPUF crosslinking since this temperature range is related to the breaking of urethane linkages [5,28]. In applications such as building flat roofing sealed with bituminous roof covering, RPUF must withstand temperatures up to 250 °C for short periods without showing adverse effects [17].

The T50% results can be associated with related to the structural decomposition of organic chains, mainly governed by cleavage of urea groups and degradation of urethane groups, which preferentially start from side chains at ~350 °C [17]. Fillers chemically linked to the polymer may influence the dehydration and evaporation of free surfactants that may be trapped inside polymer cells, including those new cell walls selectively produced by fillers, as discussed above.

All filled RPUF overcame the neat one in terms of thermal stability (T50%), which can be associated with higher molecular stiffness. Once opened cells may accelerate the thermodegradation process of RPUF [1], the disruption of some cells found in the morphological analysis did not negatively affect the RPUF thermal stability. This indicates that the filler/RPUF linkages generated the opposite effect, being capable of imparting increases in thermal stability. The TG results also indicate that the overall thermal stability of filled RPUF cannot be predicted based solely on the stability of the phases themselves. Analogous findings have been reported in the literature and again can be attributed to the filler/RPUF chemical interactions [1].

In all cases, the insertion of the fillers did not induce changes in density until 10% filler content, which can be explained by the low filler content and the similar lightweight of both RPUF and fillers (Figure 6). The densification mechanism imparted by the filler at 15% concentration can be attributed to different rheological mechanisms. Actually, incompatible fillers may induce a decrease in crosslinking density, which is accompanied by a decrease in foam expansibility [7]. This can happen for the 15% banana-filled RPUF. On the other hand, new polymer cells formed by chemically compatible fillers attached to the PU cellular structure may increase both the whole crosslinking density and foam expansibility, also leading to a more densified foam [29]. Based on above discussed morphological and thermal results, this probably occurred for the 15% bergamot-filled RPUF.

Figure 7 shows the compressive strength of the RPUF, in which there were no statistical differences attributed to the filler’s insertion. Considering the other results discussed above, the compressive results can be explained by two different opposite mechanisms. The first one is related to the formation of crosslinking between chemically compatible fillers and PU matrix. This mechanism was reported in the literature as capable of inducing gain in several mechanical properties [24,25], although some authors stated that these interactions may not be strong enough to impart increases in mechanical performance [7]. The second mechanism is detrimental in relation to the RPUF mechanical performance. This is explained by the formation of weak interphase between filler and RPUF, which is filled by moisture released from the filler and blowing gases from the RPUF rise [29]. Furthermore, the solid filler may disrupt some cell edges, releasing blowing gas and impairing the whole RPUF mechanical performance [5]. Considering that the aforementioned morphological results indicated a good filler/RPUF interaction, this later detrimental mechanism maybe had not a significant influence.

The hygroscopic performance evaluated by water uptake (Figure 8) was similar to all studied RPUF. As explained for the compressive results, the similar water uptake was possibly influenced by two opposite behaviors, which canceled each other. In [11], the authors explain the first mechanism and state that unreacted fillers may impart their hydrolytic stability to the incorporated RPUF structure. Considering that, the number of OH groups from the fruit peels probably conferred increases in RPUF hydrophilicity. On the other hand, when chemically compatible fillers are incorporated into the RPUF system, the above-discussed nucleation mechanism gives rise to new polymer bubbles, which impair the water filling, decreasing water levels [30]. In addition to that, Członka and coworkers [1] stated that morphological characteristics of the cellular structure of RPUF are the main influential factor for their hydrophobic/hydrophilic character. In this sense, as discussed above, a few morphological changes imparted by most filler configurations were really significant. In all, the similar moisture uptake can be considered a favorable result since absorbed water may have a detrimental effect on the mechanical properties of the RPUF and their thermal conductivity since liquid water has 10 times higher thermal conductivity than a typical RPUF [7].

The photographs shown in Figure 9 do not indicate a color change attributed to the filler’s insertion. Color is one of the main perceptual attributes to stimulate consumption, being important when RPUF are used as perimeter insulations, exposed parts of indoor furniture, or when glass-based laminates or other translucent materials are used as face sheets in sandwich panels [13]. According to the literature, filled RPUF normally acquires color shades from the filler [5,12,13]. All the bergamot-filled RPUF presented similar colorimetric properties if compared to the neat RPUF, which can be attributed to the similar yellowish shade of both the filler and RPUF. On the other hand, the banana peel induced decreases in both b* and C*, which means decreases in red pigments and color opacity, respectively. Once these RPUFs are clearly not red, only the increase in opacity can be considered as a real aesthetical change. Yellowish shades from both fruit peels can be attributed to some of their organic extractives, such as polyphenols, quinones, and flavonoids [1].

## 4. Conclusions

RPUF composites incorporated with different filler contents (5–15 wt%) of banana and bergamot peels were successfully manufactured. The high chemical compatibility of the fillers with the PU system was confirmed by wet chemical and FTIR results, leading to a good filler distribution indicated by SEM images. In a comparison between the fruit peels, the bergamot overperformed the banana in terms of PU host compatibility. Hygroscopic, apparent density, compressive strength, and color properties were only slightly influenced by both filler type and filler content. On the other hand, in a comparison with the neat RPUF, the incorporation of the fillers yielded increases in cell size and thermal stability of 80% and 114%, respectively. All these results are quite important since up to 15% of the oil-based RPUF was replaced by cheap and bio-based residues. The bergamot-filled RPUF incorporated at 15% stood out due to the imparted nucleating mechanisms, which promoted the formation of ellipse-like cell bubbles. This positive effect on RPUF morphology was not able to induce gains in hygroscopic and mechanical performances, which was explained by opposite mechanisms related to the disruption of some cell edges and the low strength of the filler/RPUF interface. Further studies may address measurements of thermal conductivity and dimensional stability.

## Figures and Tables

**Figure 1 polymers-14-04526-f001:**
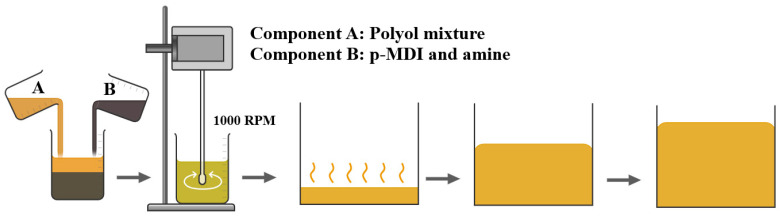
Schematic representation of the RPUF manufacturing.

**Figure 2 polymers-14-04526-f002:**
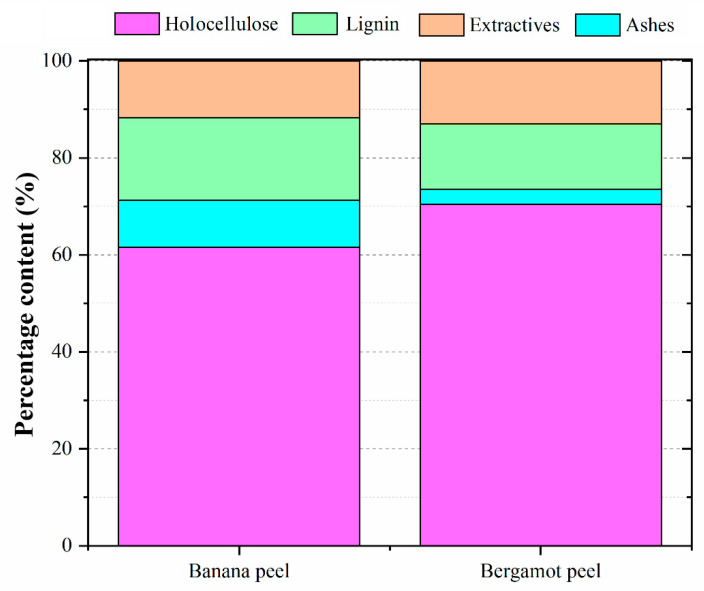
Chemical properties of the banana and bergamot peels.

**Figure 3 polymers-14-04526-f003:**
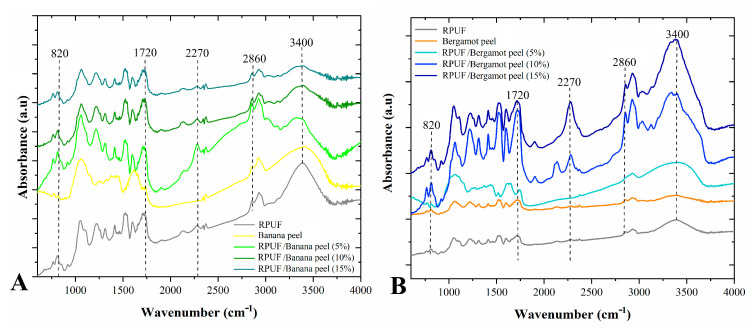
FTIR of the studied RPUF with Banana peels (**A**) and Bergamot peels (**B**). Where: RPUF is the neat rigid polyurethane foam, and filler type and filler content are the numbers after the bars and between parentheses, respectively.

**Figure 4 polymers-14-04526-f004:**
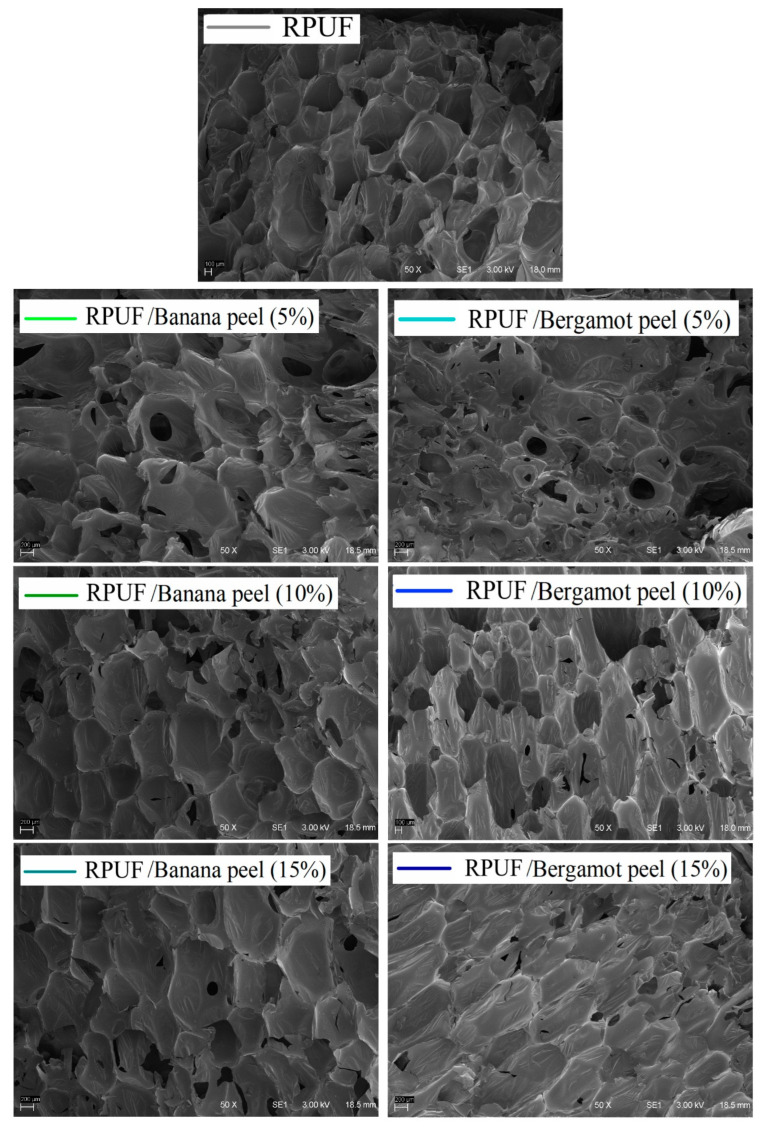
SEM images of the studied RPUF. Where: RPUF is the neat rigid polyurethane foam.

**Figure 5 polymers-14-04526-f005:**
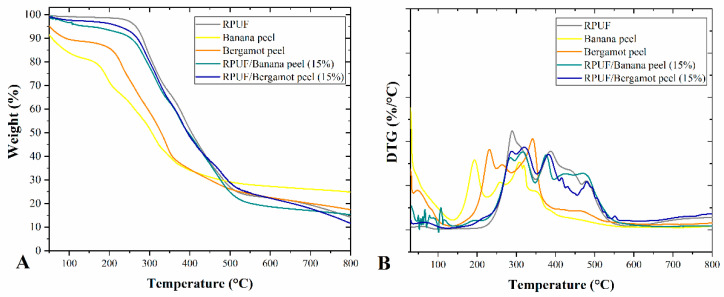
TG (**A**) and DTG (**B**) of the studied RPUF. Where: RPUF is the neat rigid polyurethane foam and filler type and filler content are the numbers after the bars and between parentheses, respectively.

**Figure 6 polymers-14-04526-f006:**
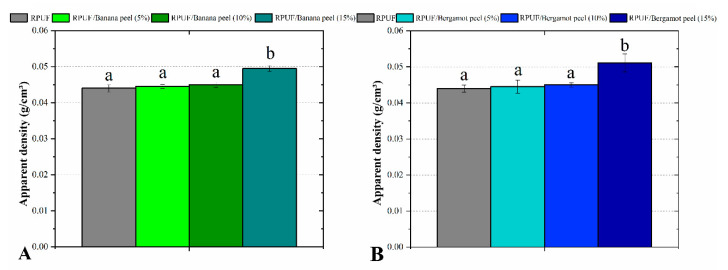
Apparent density of the studied RPUF with Banana peels (**A**) and Bergamot peels (**B**). RPUF is the neat rigid polyurethane foam and filler type, and filler content is the numbers after the bars and between parentheses, respectively. Different letters represent statistically different means.

**Figure 7 polymers-14-04526-f007:**
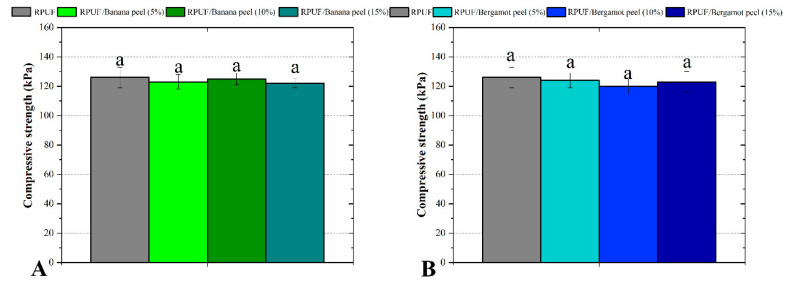
Compressive properties of the studied RPUF with Banana peels (**A**) and Bergamot peels (**B**). RPUF is the neat rigid polyurethane foam and filler type, and filler content is the numbers after the bars and between parentheses, respectively. Different letters represent statistically different means.

**Figure 8 polymers-14-04526-f008:**
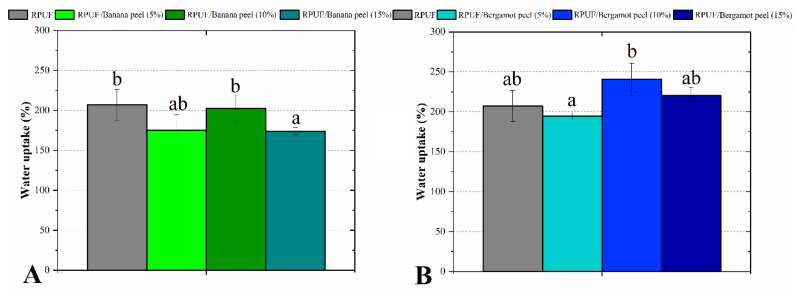
Water uptake of the studied RPUF with Banana peels (**A**) and Bergamot peels (**B**). RPUF is the neat rigid polyurethane foam and filler type, and filler content is the numbers after the bars and between parentheses, respectively. Different letters represent statistically different means.

**Figure 9 polymers-14-04526-f009:**
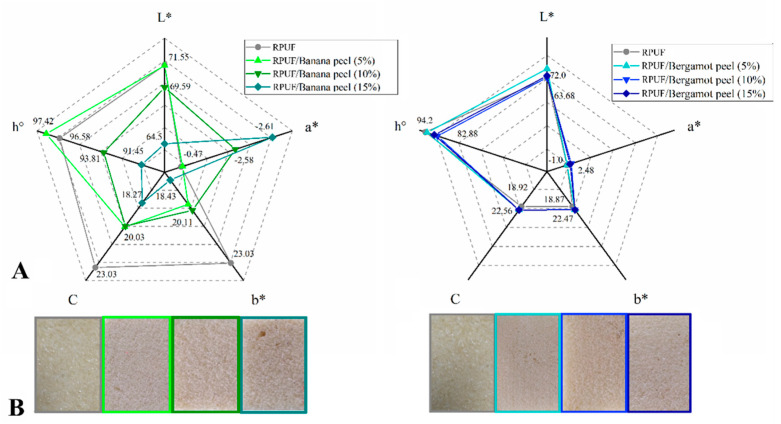
Colorimetric properties (**A**) and photographs (**B**) of the studied RPUF with Banana peels (**A**) and Bergamot peels (**B**). RPUF is the neat rigid polyurethane foam and filler type, and filler content is the numbers after the bars and between parentheses, respectively. Where: L* is brightness; a* is green-red coordinate; b* is blue-yellow coordinate; C is chroma; h° is hue angle.

**Table 1 polymers-14-04526-t001:** Morphological properties of the studied RPUF.

Group	Cell Length (µm)	Cell Width (µm)	Anisotropy Index
RPUF	630.12 ^(221.12 ab)^	615.16 ^(173.19 b)^	1.01 ^(0.095 a)^
RPUF/Banana peel (5%)	569.14 ^(260.09 ab)^	516.98 ^(206.09 ab)^	1.09 ^(0.21 a)^
RPUF/Banana peel (10%)	994.45 ^(238.90 bc)^	597.09 ^(201.89 ab)^	1.72 ^(0.28 b)^
RPUF/Banana peel (15%)	956.87 ^(286.76 bc)^	637.98 ^(183.76 b)^	1.54 ^(0.40 b)^
RPUF/Bergamot peel (5%)	340.10 ^(40.19 a)^	380.01 ^(59.67 ab)^	0.90 ^(0.11 a)^
RPUF/Bergamot peel (10%)	653.01 ^(116.38 ab)^	297.92 ^(73.45 a)^	2.26 ^(0.44 c)^
RPUF/Bergamot peel (15%)	1138.98 ^(339.83 c)^	527.56 ^(137.01 ab)^	2.14 ^(0.10 c)^

Where: RPUF is the neat rigid polyurethane foam, and filler type and filler content are the numbers after the bars and between parentheses, respectively. Different letters represent statistically different means.

**Table 2 polymers-14-04526-t002:** Main thermal events evaluated by TG analysis of the studied RPUF.

Group	T_2%_	T_5%_	T_50%_
RPUF	70.12	88.14	237.41
Banana peel	72.01	91.21	318.12
RPUF/Banana peel (5%)	100.25	232.11	492.37
RPUF/Banana peel (10%)	86.93	226.81	497.67
RPUF/Banana peel (15%)	87.10	197.74	508.10
Bergamot peel	70.86	93.94	295.21
RPUF/Bergamot peel (5%)	80.09	109.16	312.48
RPUF/Bergamot peel (10%)	73.25	95.76	302.56
PU/Bergamot peel (15%)	87.45	217.20	393.45

Where: RPUF is the neat rigid polyurethane foam, and filler type and filler content are the numbers after the bars and between parentheses, respectively.

## Data Availability

The study did not report any data.

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
