# Peer review of "Rigid Polyurethane Biofoams Filled with Chemically Compatible Fruit Peels"

_polymers, 2022, doi:10.3390/polym14214526_

Round 1

Reviewer 1 Report

The manuscript is mainly about the properties of RPUF composites using polysaccharides-rich fruit peels as fillers, including morphological, thermal, mechanical, hygroscopic and colorimetric properties. Some of the problems found are as follows:

1. In the TG results, the char residues of RPUF and Bergamot peel at 800 ℃ is about 15% and 18%, respectively, while that of RPUF/Bergamot pee (15%) at about 455 ℃ is 0, and it is still losing weight with the increase of temperature. Therefore, the result of RPUF/Bergamot pee is unreasonable and needs to be retested.

2. In the TG results, Bergamot peel and Banana peel continuously lose weight from 25 ℃, which is probably caused by the volatilization of water. Why not dry the samples before testing?

3. I am confused with "The TG results shown in Table 2 and Figure 5 confirm FTIR results and again corroborates that all the studied RPUF have a similar chemical composition", please explain it in detail.

4. Please add information about p-MDI, PEG-400, silicon oil, and dimethylbenzylamine in "2.1. materials " to improve the reproducibility of the experiment.

5. English needs improving.

Author Response

REBUTTAL TO REFEREES' REMARKS

The authors would like to thank the reviewers for their valuable contribution to the manuscript. Our answers are listed below. All suggestions were accepted and the manuscript was revised accordingly. The changes were highlighted in blue colour.

Reviewer 1

Comments to the Author

English language and style are fine/minor spell check required. English needs improving.

Answer: The text was fully revised and scattered errors were corrected accordingly.

In the TG results, the char residues of RPUF and Bergamot peel at 800 ℃ is about 15% and 18%, respectively, while that of RPUF/Bergamot pee (15%) at about 455 ℃ is 0, and it is still losing weight with the increase of temperature. Therefore, the result of RPUF/Bergamot pee is unreasonable and needs to be retested.

Answer: Thank you for the comment, we ran again this analysis. Please, look at Figure 5 and Table 2.

In the TG results, Bergamot peel and Banana peel continuously lose weight from 25 ℃, which is probably caused by the volatilization of water. Why not dry the samples before testing?

Answer: We agree that this initial mass loss is attributed to a drying process. However, a new analysis for a pre-dried sample won’t bring a different or relevant result.

I am confused with "The TG results shown in Table 2 and Figure 5 confirm FTIR results and again corroborates that all the studied RPUF have a similar chemical composition", please explain it in detail.

Answer: Thank you for your comment. We deleted this sentence to avoid any misunderstanding.

Please add information about p-MDI, PEG-400, silicon oil, and dimethylbenzylamine in "2.1. materials " to improve the reproducibility of the experiment.

Answer: Done as suggested. Please, see at the first paragraph of the section 2.1.

Reviewer 2 Report

Dear Authors,

I studied your manuscript entitled "Rigid polyurethane biofoams filled with chemically compatible fruit peels". Some spaces need to be improved in terms of journal quality. I recommend a major revision before further consideration for publication in the Polymers.

1) The quality of the abstract and conclusion should be enhanced by the inclusion of significant research findings. More quantitative data in these sections would be beneficial.

2) The recent literature review on the Rigid polyurethane biofoams from different authors and years should be summarized in a table for benchmarking purposes and discussed in detail with your research findings.

3) How did you select the curing temperature and time?

4) Please describe the mixing procedure in detail. Did you employ a high-shear mixer and/or ultrasonic homogenizer?

5) Picture/Table captions could provide more details, so that readers will not have to go back to the experimental section for necessary details.

6) Please provide manufacturer details (model, city, or country) for all characterization instruments.

7) It is strongly suggested to provide the results of thermal conductivity and dimensional stability.

8) The manuscript needs to be thoroughly revised because it contains a few typos and errors.

Author Response

Reviewer 2

The quality of the abstract and conclusion should be enhanced by the inclusion of significant research findings. More quantitative data in these sections would be beneficial.

Answer: Done. Please, see at these sections.

The recent literature review on the Rigid polyurethane biofoams from different authors and years should be summarized in a table for benchmarking purposes and discussed in detail with your research findings.

Answer: we think that the fourth paragraph of the Introduction cites several studies on RPUF incorporated with plant-derived fillers, in addition to the main results reported in these articles. Our results were also compared with similar ones described in previous studies. Therefore, it is not clear to us the need for a new table with this information already contained in the manuscript.

How did you select the curing temperature and time?

Answer: These procedures were performed according to previous studies of the group. This was explained in the text. Please, look at the last paragraph before the Figure 1.

Please describe the mixing procedure in detail. Did you employ a high-shear mixer and/or ultrasonic homogenizer?

Answer: Done. Please, see at paragraph before Figure 1.

Picture/Table captions could provide more details, so that readers will not have to go back to the experimental section for necessary details.

Answer: Done. Please, see at the Figure/Table captions.

Please provide manufacturer details (model, city, or country) for all characterization instruments.

Answer: Done. Please, see at the section 2.2.

It is strongly suggested to provide the results of thermal conductivity and dimensional stability.

Answer: We are sorry, but we do not have enough samples for these tests. However, this may be included in further studies as we mentioned in the new version of our conclusion.

The manuscript needs to be thoroughly revised because it contains a few typos and errors.

Answer: The text was fully revised and scattered errors were corrected accordingly.

Round 2

Reviewer 1 Report

This paper can be accepted.

Reviewer 2 Report

Dear Authors,

I have recommended the publication of your article as is.